# Structure of *Klebsiella pneumoniae* adenosine monophosphate nucleosidase

**Brian C. Richardson[1], Roger Shek[2], Wesley C. Van Voorhis[2], Jarrod B. French**[1]*

**1** The Hormel Institute, University of Minnesota, Austin, Minnesota, United States of America, **2** Division of Allergy and Infectious Diseases, Department of Medicine, Center for Emerging and Re-emerging Infectious Diseases, University of Washington School of Medicine, Seattle, WA, United States of America

* jfrench@umn.edu

**Data Availability Statement:** Structural data are available under PDB code 7UWQ and EMBD code EMD-26838.

**Funding:** This project has been funded in part with Federal funds from the National Institute of General

## Abstract

*Klebsiella pneumoniae* is a bacterial pathogen that is increasingly responsible for hospital-acquired pneumonia and sepsis. Progressive development of antibiotic resistance has led to higher mortality rates and creates a need for novel treatments. Because of the essential role that nucleotides play in many bacterial processes, enzymes involved in purine and pyrimidine metabolism and transport are ideal targets for the development of novel antibiotics. Herein we describe the structure of *K. pneumoniae* adenosine monophosphate nucleosidase (KpAmn), a purine salvage enzyme unique to bacteria, as determined by cryoelectron microscopy. The data detail a well conserved fold with a hexameric overall structure and clear density for the putative active site residues. Comparison to the crystal structures of homologous prokaryotic proteins confirms the presence of many of the conserved structural features of this protein yet reveals differences in distal loops in the absence of crystal contacts. This first cryo-EM structure of an Amn enzyme provides a basis for future structure-guided drug development and extends the accuracy of structural characterization of this family of proteins beyond this clinically relevant organism.

## Introduction

The gram-negative bacterium *Klebsiella pneumoniae* is an opportunistic pathogen. Natively found on human mucosal surfaces, it is a frequent cause of nosocomial pneumonia and sepsis due in large part to its protective capsule and ability to form robust biofilms on medical equipment [1]. In the past two decades, *K. pneumoniae* has garnered greater concern due to its increasing resistance to beta-lactam antibiotics, including the ability to inactivate carbapenems typically resistant to extended-spectrum beta-lactamases [2]. As such, it is recognized as one of the six pathogens of greatest concern in nosocomial infection known as the ESKAPE pathogens after their genera: *Enterococcus faecium*, *Staphylococcus aureus*, *Klebsiella pneumoniae*, *Acetobacter baumannii*, *Pseudomonas aeruginosa*, and *Enterobacter* spp. [3]. Thus, as with *Pseudomonas aeruginosa* [4] and *Staphylococcus aureus* [5] among the other ESKAPE pathogens, the need for novel antibiotics targeting *K. pneumoniae* is pressing.

The ideal antibiotic is highly deleterious to pathogenic bacteria but lacking in similar effects on the host cells. For instance, beta-lactams act on bacterial peptidoglycans [6,7], a structure

Medical Science and National Institute of Allergy and Infectious Diseases, National Institutes of Health, Department of Health and Human Services, under grant number R35GM124898 and contract No. HHSN272201700059C, respectively. The funding agencies that supported this work had no role in study design, data collection and analysis, decision to publish, or preparation of the manuscript.

**Competing interests:** The authors have declared that no competing interests exist.

**Fig 1. *K. pneumoniae* AMP nucleosidase chemistry.** In *K. pneumoniae*, AMP is broken down into adenine and ribose 5-phosphate by the enzyme AMP nucleosidase (Amn). In many other organisms, including mammals, AMP is deaminated by AMP deaminase (AD) to generate inosine monophosphate.

not found in eukaryotes, and are better tolerated than antibiotics such as chloramphenicol which can cross-react with mitochondrial protein synthesis as well as their target bacterial ribosomes [8,9]. The preferred targets of novel antibiotics are therefore pathways unique to bacteria.

Biochemical studies of nucleotide processing pathways in *Azotobacter vinelandii* and *Escherichia coli* (*E. coli*) identified a key difference between prokaryotes and eukaryotes in their regulation of the levels of adenosine monophosphate (AMP), an essential component of both RNA synthesis and energy storage [10]. Whereas eukaryotes can deaminate AMP to inosine monophosphate, prokaryotes cleave the adenine base from the ribose 5-phosphate moiety *via* AMP nucleosidase (Amn, EC 3.2.2.4), additionally salvaging adenine (Fig 1). Adenosine analogs including formycins and pyrazofurins demonstrate strong antibiotic ability, but also cross-react considerably with mammalian biology [11,12]; structure-based modification for specificity against prokaryotic reactions, including that of Amn, has been proposed as a means of developing these agents into viable therapeutic treatments [13,14].

Toward this goal, crystal structures have been determined of both *E. coli* [14] and *Salmonella typhimurium* [15] Amn. However, development of antibiotics toward a broad spectrum of infectious agents benefits from a similarly broad assessment of variability in the target molecular structures. Furthermore, the remarkable power of X-ray crystallography in determining protein structure can be limited by the distortions caused by crystal contacts [16], as well as the non-native solution conditions typically required to crystallize and cryoprotect the proteins [17]. Using cryo-EM, the structure of *K. pneumoniae* Amn was determined independent of these protein crystallization considerations [18].

## Results

### Cryo-EM structure of *K. pneumoniae* Amn

The cryo-EM map of *K. pneumoniae* Amn was determined from images collected on a Titan Krios equipped with a Falcon III detector to an estimated resolution of 3.03 Å (Fig 2C). *Ab initio* models confirmed the D3 symmetry expected from the homologous crystal structures, and all further refinement and model building enforced this symmetry in the interest of improving resolution and mitigating the moderate orientation bias of the sample (Fig 2D). The resulting

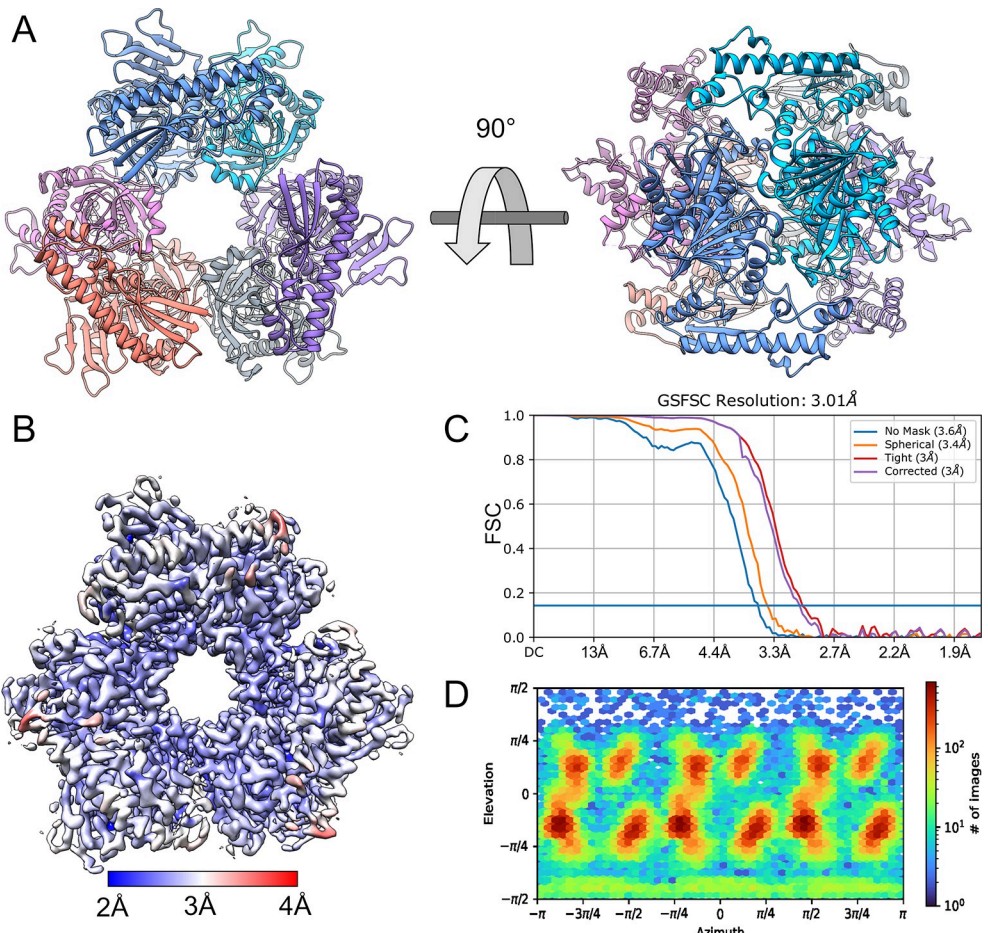

**Fig 2. Structure of _K. pneumoniae_ Amn.** KpAmn has an overall hexameric structure with D3 symmetry (A, shown colored by chain). The sharpened electron density map (B), colored by local resolution (CryoSPARC at FSC = 0.143 and contoured at 3σ = 0.604V) reveals the well-defined features of the secondary structure. The resolution of the structure was calculated to be 3.01 Å using the gold standard Fourier shell correlation (GFSC, C). The particles used to generate the final model had a reasonably broad angular distribution (D).

map proved suitable for modeling the protein, and KpAmn was built into it using the homologous crystal structures as an initial model followed by manual rebuilding (Fig 2A). The map lacks density for the first seven residues of the protein, as well as presumptive loops at residues 61–79, 156–166, 362–365, and 468–478, but otherwise possesses the features expected at the estimated resolution, including clear density for side chains (S1 Fig). The protein comprises a well-resolved C-terminal hexameric core consisting of an extensive β-sheet structure surrounded by α-helices, and a somewhat less ordered ancillary N-terminal domain, dominated by a long α-helix,forming trimeric layers on each side of the core (Fig 2A and 2B).

In keeping with the prior _E. coli_ structures, no significant ligand density is evident at the active sites. Comparison to the presumed active site that binds the formycin inhibitor in PDB entry 1T8S shows similar side chain orientations in the KpAmn structure and the corresponding PDB entry 1T8R _apo_ structure (Fig 3). The active site is composed of residues from two chains; a cleft of conserved residues in one chain binds the nucleobase and sugar of the adenosine analog, with H188 and Y189 of the apposed chain contributing to stabilize the charged phosphate group. In the _apo_ structure as determined, the core residues expected to stabilize the nucleobase and sugar (N205, W383, M404, E405, and D428) are well positioned to do so,

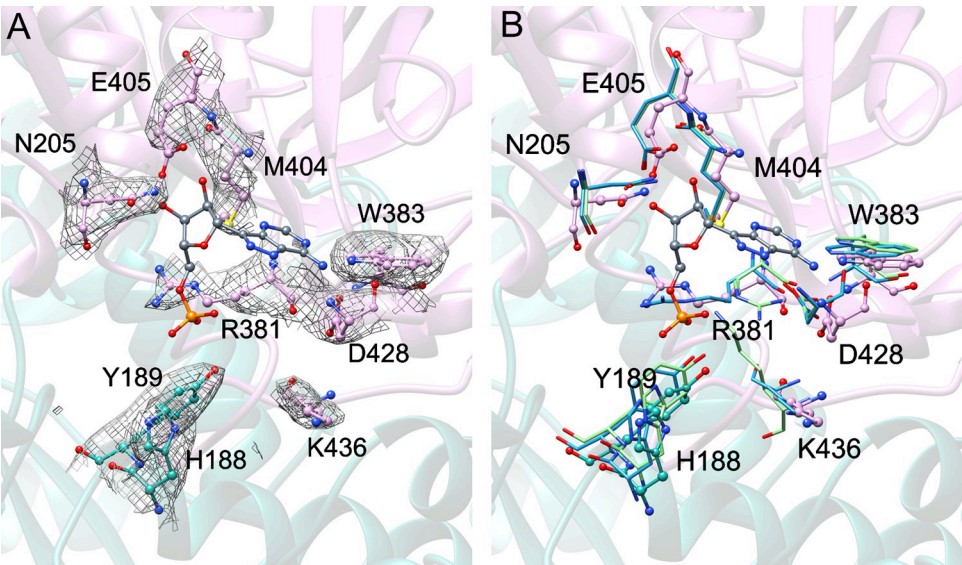

**Fig 3. Active site of KpAmn.** Presumed active site of KpAmn based on superposition with *E. coli* Amn (Protein Data Bank (PDB) entry 1T8S). Residues from two chains (A and B) make up the active site and are colored with green carbon atoms and yellow carbon atoms, respectively. The inhibitor formycin is shown with cyan carbon atoms and was positioned by superimposing the structure of PDB entry 1T8S over the solved structure of KpAmn. In these ball-and-stick models, oxygen atoms are shown in red, nitrogen atoms are shown in blue, sulfur atoms are shown in yellow, and the phosphate atoms are shown in orange. In panel A, the electron density is from a sharpened map and is contoured at 3σ. In panel B, the corresponding *E. coli* residues are show as sticks only in teal (without ligand) and green (with ligand).

whereas R381 and K436 are in alternative conformations in the absence of the interacting phosphate. The surrounding residues are well resolved; weak density outward from the active site is interpreted as residues 435–443, possibly stabilizing the *apo* protein while remaining flexible enough to permit AMP access.

## Comparison to Amn homologs

As expected, a structure-based search of the PDB using the DALI server [19] identified the *E. coli* Amn structure family alongside weaker hits for other purine-binding enzymes (Table 1). The crystal structures of the *E. coli* Amn homolog have been determined with and without substrates bound [14], while the unpublished structure of *Salmonella typhimurium* Amn (PDB entry 2GUW) [15] was determined only in an unliganded state. Several other, considerably more distant sequence homologs that lack the N-terminal domain are also available in the PDB (Table 1); consistent with the variation in annotated function, conservation in the Amn active site is limited to residues M404, E405, and D428 interacting with the purine base and sugar (S2 Fig). The cryo-EM structure of *K. pneumoniae* Amn primarily differs from the *apo* homologs in its loop regions away from the active site (Fig 4A), either by lacking density for the loop entirely or being in a clearly different conformation. As these loops are those that form presumptively physiologically irrelevant crystal contacts in the crystal structure, the cryo-EM model of K. pneumoniae Amn, lacking such interactions, is likely to better reflect the protein's conformation in solution.

A broader comparison of *K. pneumoniae* Amn to its homologs can be assessed by considering sequence conservation without restriction to those with known structures. Scoring by Con-SURF [20] indicates the expected strong conservation at the active site and the protein assembly surfaces, with more variability observed in residues on the outside of the complex

**Table 1. Structure comparison results.** Pairwise structure comparison of KpAmn to all structures in the Protein Data Bank as of March 2022 (the top 10 of 43 total results are given). Further alignment details are provided in S2 Fig.

| PDB ID[a] | Z-score[b] | RMSD[c] | No. Res.[d] | % ID[e] | Description | Species |
|-----------|-----------|---------|-------------|---------|-------------|---------|
| 1T8R | 46.9 | 1.2 | 463 | 90 | Amp Nucleosidase | *Escherichia coli* |
| 4LDN | 24.2 | 2.8 | 244 | 12 | Purine Nucleoside Phosphorylase | *Aliivibrio fischeri* |
| 3MB8 | 23.7 | 2.7 | 239 | 20 | Purine Nucleoside Phosphorylase | *Toxoplasma gondii* |
| 3QPB | 22.9 | 2.6 | 231 | 18 | Uridine Phosphorylase | *Streptococcus pyogenes* |
| 3BJE | 22.2 | 2.6 | 239 | 13 | Putative Nucleoside Phosphorylase | *Trypanosoma brucei* |
| 2QSU | 17.9 | 2.6 | 208 | 18 | 5′-Methylthioadenosine Nucleosidase | *Arabidopsis thaliana* |
| 4QAS | 13.3 | 2.9 | 176 | 10 | CT263 | *Chlamydia trachomatis* |
| 4PR3 | 13.0 | 3.0 | 174 | 16 | 5′-Methylthioadenosine Nucleosidase | *Brucella melitensis* |
| 2GFQ | 10.3 | 3.1 | 166 | 8 | UPF0204 Protein PH00006 | *Pyrococcus horikoshii* |
| 3VR0 | 10.2 | 3.4 | 163 | 8 | Uncharacterized Protein | *Pyrococcus furiosus* |

[a] Protein Data Bank Identifier.

[b] The calculated Z-score for the alignment [19].

[c] Root means square deviation for the alignment.

[d] Number of residues in structure being compared.

[e] Percentage sequence identity.

and in the N-terminal domain overall (Figs 4B and S3 Fig). The loops distorted by crystal contacts are not well conserved. Notably, the loop covering the active site is strongly conserved, suggesting an important role in protein activity, ligand gating or structural stabilization, despite its weak density.

## Catalytic activity of *K. pneumoniae* Amn

As a final confirmation that the purified KpAmn was correctly folded and that its sequence-based identification was correct, the protein's catalytic activity was characterized spectrophotometrically using a coupled assay (Fig 5). The determined $K_M$ of 390 μM and $V_{max}$ of 50 nmol/min/mg are comparable to those previously reported for *E. coli* (120 μM and 16 nmol/min/mg) and *A. vinelandii* (80 μM and 25 nmol/min/mg) homologs [21–24].

## Discussion

Among the most common criticisms of structures determined by X-ray crystallography is that their conformation in the crystal lattice may not reflect their conformation in solution, and indeed significant differences have been observed in multidomain proteins when analyzed by solution techniques such as nuclear magnetic resonance or small-angle X-ray scattering [16,25–27]. Conversely, crystallography remains the standard for the high-resolution atomic structures required for analysis of small-molecule ligand binding despite the recent progress made in cryo-EM technology, and these differences in structure are frequently restricted to less conserved surface sequences [28]. In either case, the full determination of a protein's range of conformations beyond modeling *via* B-factors greatly benefits from the use of multiple complementary techniques.

Comparing the cryo-EM structure of KpAmn to the crystal structures of *E. coli* Amn provides such a determination for this enzyme. This high level of structural conservation in the active site implies that inhibitors that target Amn enzymes may have broad applicability across species. The KpAMn structure, and analysis herein, provides a foundational resource for the structure-guided development of such Amn-targeted therapies.

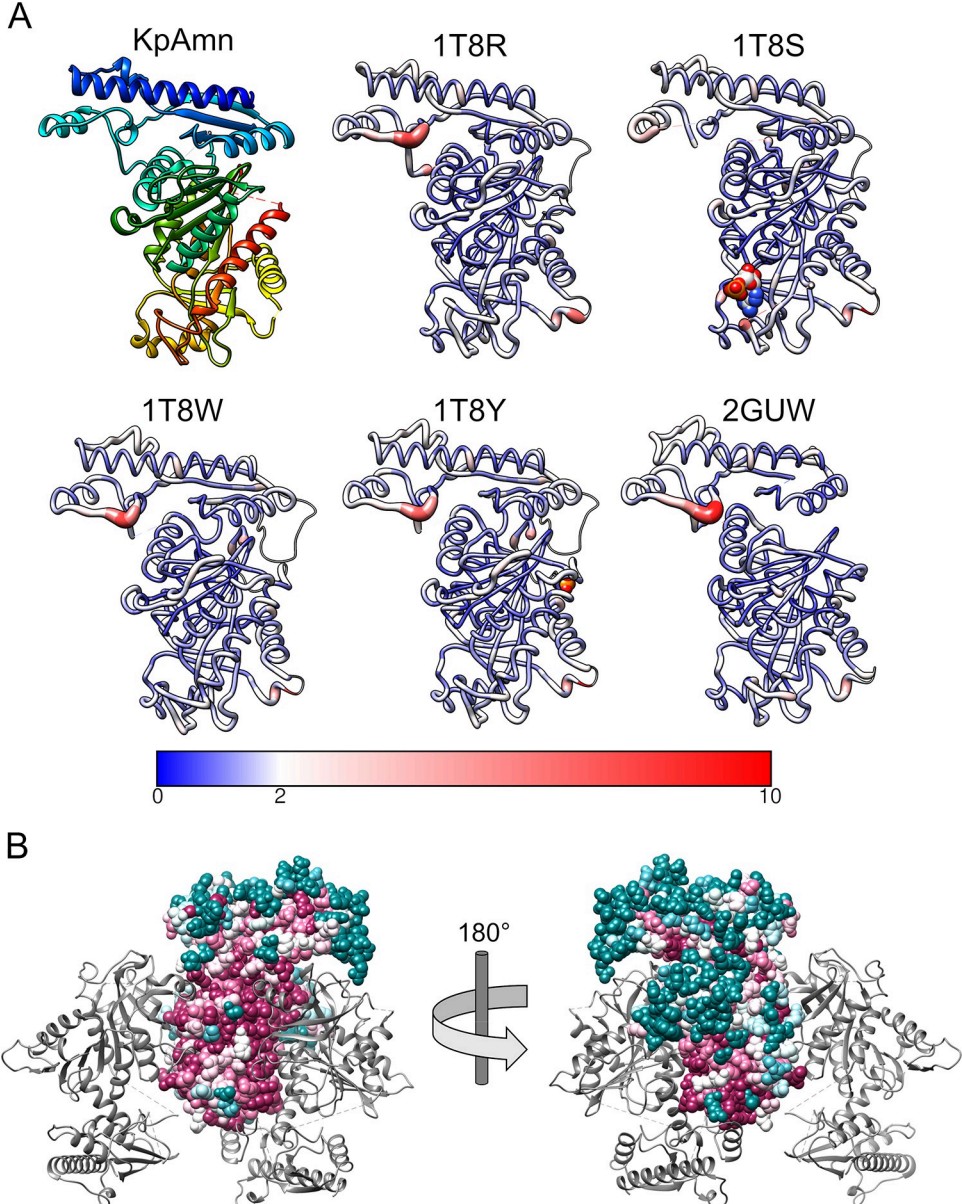

**Fig 4. Comparison of Amn between species.** All PDB structures identified by BLAST as close Amn homologs were aligned to Amn in UCSF Chimera (A). Color and ribbon width indicate the all-atom spatial distance RMSD of each residue from that of KpAmn; a thin line represents absence of the corresponding loop. Ligands bound by PDB entries 1T8R and 1T8Y are shown as spheres. ConSurf-derived conservation of Amn (B), ranging from low (teal) to high (magenta). The alignment is detailed in S3 Fig.

More generally, the determination of the KpAmn structure addresses limitations of the prior structures with respect to their utility in systematic analysis, including structural validation and prediction tools. The *E. coli* structures, though certainly correct, pre-date the advent of mandatory structure factor submission and therefore cannot be compared to the experimental electron density map. Conversely, while the *S. typhimurium* structure does not have such an issue, the structural statistics suggest that the model as submitted may benefit from further refinement.

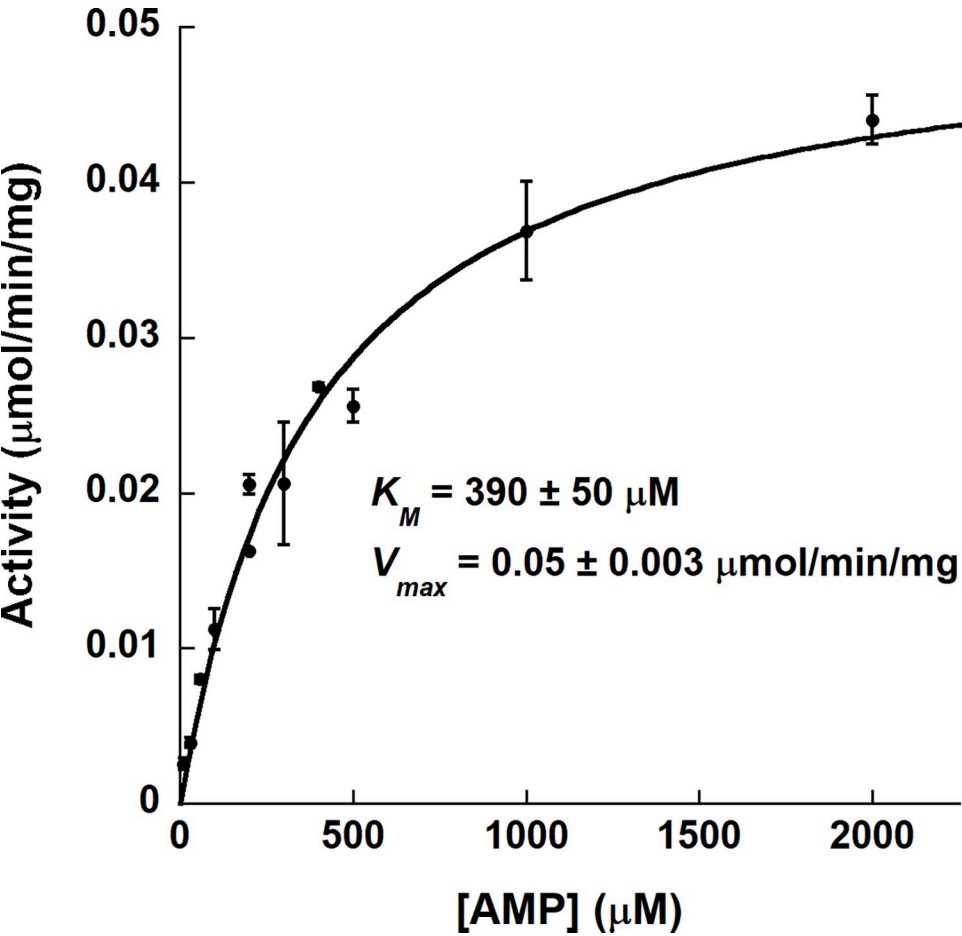

**Fig 5. Catalytic activity of *K. pneumoniae* Amn.** The activity of KpAmn was measured by quantifying the generation of the product, adenine. The initial rate data at varying concentrations of AMP was fit by the Michaelis-Menten equation to yield the kinetic constants shown.

In conclusion, the determination of the first cryo-EM structure of an Amn protein provides a refinement of its unperturbed structure, as well as a model of its molecular structure of potential relevance to the antibiotic arms race underway against *K. pneumoniae*.

## Materials and methods

### Expression and purification of K. pneumoniae Amn

KpAmn was cloned, expressed, and purified using standard SSGCID protocols. Briefly, the gene was cloned into a pET-14b derived vector and expressed in E. coli BL-21(DE3) R3 Rosetta cells in 2 L autoinduction media using a LEX Bioreactor. The culture was harvested and stored at -80°C until purification. The cell pellet was resuspended in buffer containing 25 mM HEPES pH 7.0, 500 mM NaCl, 5% glycerol, 30 mM imidazole, 0.025% sodium azide, 0.5% CHAPS, 10 mM MgCl2, 1 mM TCEP, 250 μg/mL AEBSF, 0.05 μg/mL lysozyme, and 25 U/mL benzonase, lysed by sonication, and then clarified by centrifugation at 26000 g for 45 min at 4°C. The soluble supernatant was loaded onto a HisTrap FF 5 mL (GE Healthcare, New Jersey, USA), washed with wash buffer (25 mM HEPES pH 7.0, 500 mM NaCl, 5% glycerol, 30 mM imidazole, 0.025% sodium azide) and eluted from the column by wash buffer supplemented

with 350 mM imidazole. The eluted protein was further purified by gel-filtration on a Superdex 200 26/600 size-exclusion chromatography column (GE Healthcare) in 25 mM HEPES pH 7.0, 500 mM NaCl, 5% Glycerol, 2 mM DTT, and 0.025% sodium azide. Fractions were visualized using SDS-PAGE and the fractions containing the purified protein were pooled and concentrated using Amicon Ultra centrifugal filters, flash frozen in liquid nitrogen, and stored at -80˚C.

### Cryo-EM data collection and molecular modeling of K. pneumoniae Amn

3 µL of purified KpAmn was applied to a 300 mesh copper C-flat grid with a hole size of 1.2 µm and hole spacing of 1.3 µm (Electron Microscopy Sciences), blotted and vitrified with a Vitrobot Mark IV (Thermo Fisher) with a blotting time of 6 s and blotting force of 0 under >90% humidity at 4˚C. Images were collected on a 300 kV Titan Krios equipped with a Falcon 3EC direct electron detector. Automated data collection was carried out using the EPU 2.8.01256REL software (Thermo Fisher Scientific) at a nominal magnification of 96000 x corresponding to a calibrated pixel size of 0.8891 Å with a defocus range from 1.0 to 2.6 µm. Image stacks comprising 45 frames were recorded at an estimated total dose of 60 electrons/ Å$^2$.

An initial 3D map was determined in RELION [29] by picking ~1 million particles from 2237 motion- and CTF-corrected micrographs by Laplacian of Gaussian, then removing junk particles with several rounds of 2D classification. Following ab initio 3D model construction and further refinement, the resulting 3.5 Å map was used to create a template for particle picking in cryoSPARC [30].

To obtain the best final map in cryoSPARC, ~1.5 million particles were picked and extracted in 300x300 pixel boxes, and filtered by 2D classification first after downscaling to 72x72 resolution and then at full resolution to yield 566,674 final particles. These were used to refine the input map in D3 symmetry to a gold-standard FSC = 0.173 estimated resolution of 3.08Å. Applying a sharpening B-factor of -160Å$^2$ produced the final map used for refinement.

Working from an initial model of EcAmn (PDB code 1T8R) mutated appropriately and fit into the sharpened cryo-EM map as a rigid body, adjustments were made in Coot [31] and refined in PHENIX [32], with enforced symmetry. Visualization of the resulting structure was performed in UCSF Chimera [33]. Data collection and refinement parameters are summarized in Table 2.

### Nucleosidase activity assay

The activity of KpAmn was measured spectrophotometrically by determining the rate of production of the product, adenine. The production of adenine was quantified by using an established adenine deaminase assay [36], in which the deamination of adenine is coupled to the NADPH-dependent production of glutamate by the enzyme glutamate dehydrogenase. The assay was run in a 96-well plate (100 µL total volume) at 25˚C in 20 mM HEPES, pH 7.5, 100 mM NaCl, 0.4 mM NADH, 1 mM α-ketoglutarate, 1 mM MnCl$_2$, 4 units of glutamate dehydrogenase (Sigma), varying concentrations of AMP, and 10 µg of E. coli adenine deaminase [36]. Control reactions were conducted with ammonium sulfate and adenine, prior to measuring KpAmn kinetics, to ensure that neither the glutamate dehydrogenase-catalyzed reaction nor the deaminase reaction, respectively, were rate limiting. The reaction was initiated by adding 20 µg of KpAmn and monitored at 340 nm. The initial rate, before the reaction had reached 10% completion, was calculated and plotted against the AMP concentration. A fit to this data, using the Michaelis-Menten equation, was employed to derive the kinetic parameters of the enzyme.

**Table 2. Cryo-EM data collection and refinement parameters.**

| | |
|---|---|
| PDB code | 7UWQ |
| EMD code | EMD-26838 |
| Number of grids used | 1 |
| Grid type | C-flat gold |
| Microscope | Titan Krios |
| Detector | Falcon III |
| Voltage (kV) | 300 |
| Magnification | 96000 |
| Spherical aberration (mm) | 2.7 |
| Exposure ($e^-/\text{Å}^2$) | 60 |
| Defocus (μm) | -1.0 to -2.6 |
| Pixel size (Å) | 0.8891 |
| Frames/movie | 40 |
| Number of micrographs used | 679 |
| Number of particles used | 114781 |
| Map resolution (Å) (.143 FSC) | 3.01 |
| Map sharpening B-factor ($\text{Å}^2$) | -100 |
| Mask-model CC | 0.86 |
| Non-hydrogen atoms | 20538 |
| Protein residues | 2598 |
| Bond length RMSD (Å) | 0.005 |
| Bond angle RMSD (Å) | 0.559 |
| MolProbity score [34] | 1.30 |
| Clash score | 3.23 |
| Ramachandran outliers (%) | 0 |
| Ramachandran allowed (%) | 3.07 |
| Ramachandran favored (%) | 96.93 |
| CaBLAM outliers (%) | 0.73 |
| Rotamer outliers (%) | 0 |
| EMRinger score [35] | 3.23 |

# Supporting information

**S1 Fig. Model quality assessment.** Three randomly selected motion-corrected micrographs with suitable CTF resolution estimates are shown (A). The final density map was obtained from particles in the 2D classes identified by a green border (B). Clear and appropriate density for side chains was evident in both lower (C, residues 8–38) and higher (D, residues 284–294) resolution regions; density is contoured to just above the local noise threshold within 2.4Å of any displayed atom. No KpAmn (blue model) density is observed for the residues corresponding to loops implicated in the *E. coli* structures' crystal contacts (green and magenta) (E). (TIF)

**S2 Fig. Alignment of KpAmn to known protein structures.** Structural matches to KpAmn identified by DALI were aligned using MUSCLE [37], generating a phylogenetic tree (A) and residue alignment (B). Active site residues are marked with boxes. (TIF)

**S3 Fig. Conservation of KpAmn by residue.** The 150-sequence alignment generated by ConSurf is represented in logo format using WebLogo [38], in which letter height corresponds to

strength of individual residue conservation. The N-terminal unresolved residues and C-terminal five residues following a series of unresolved residues are omitted. Active site residues are marked with boxes.
(TIF)

## Acknowledgments

We thank Janarjan Bhandari for cryo-EM data collection. We would like to thank the entire SSGCID team for their support, especially the cloning and protein-production group at the University of Washington.

## Author Contributions

**Data curation:** Brian C. Richardson.

**Funding acquisition:** Wesley C. Van Voorhis, Jarrod B. French.

**Investigation:** Brian C. Richardson, Roger Shek.

**Project administration:** Jarrod B. French.

**Supervision:** Wesley C. Van Voorhis, Jarrod B. French.

**Validation:** Brian C. Richardson, Jarrod B. French.

**Writing – original draft:** Brian C. Richardson.

**Writing – review & editing:** Brian C. Richardson, Roger Shek, Wesley C. Van Voorhis, Jarrod B. French.

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
