## [Decision Letter · Decision Letter 0]

24 Jun 2022

PONE-D-22-14060Structure of Klebsiella Pneumoniae Adenosine Monophosphate NucleosidasePLOS ONE

Dear Dr. French,

Thank you for submitting your manuscript to PLOS ONE. After careful consideration, we feel that it has merit but does not fully meet PLOS ONE’s publication criteria as it currently stands. Therefore, we invite you to submit a revised version of the manuscript that addresses the points raised during the review process.

Manuscript has been seen by 4 reviewers, two with cryoEM expertise and two being subject experts. Please address all the concerns.

We look forward to receiving your revised manuscript.

Kind regards,

Janesh Kumar, Oh.D.

Academic Editor

PLOS ONE

Journal Requirements:

Additional Editor Comments:

Please address all the concerns raised by the reviewers.

Reviewers' comments:

Reviewer's Responses to Questions

**Comments to the Author**

1. Is the manuscript technically sound, and do the data support the conclusions?

Reviewer #1: Partly

Reviewer #2: Partly

Reviewer #3: Partly

Reviewer #4: Yes

2. Has the statistical analysis been performed appropriately and rigorously? 

Reviewer #1: Yes

Reviewer #2: No

Reviewer #3: I Don't Know

Reviewer #4: Yes

3. Have the authors made all data underlying the findings in their manuscript fully available?

Reviewer #1: No

Reviewer #2: No

Reviewer #3: Yes

Reviewer #4: Yes

4. Is the manuscript presented in an intelligible fashion and written in standard English?

Reviewer #1: Yes

Reviewer #2: Yes

Reviewer #3: Yes

Reviewer #4: Yes

5. Review Comments to the Author

Reviewer #1: The article reports the first cryo-EM structure of an Amn enzyme except no additional structural information is provided by the authors. The authors did not provide any CryoEM micrographs, images, 2D classification images or 3D reconstruction details.

As described by authors the EM structure is not providing any new information regarding the KpAmn structure only physiologically irrelevant different loop conformations due crystal contacts in the crystal structure.

The authors either rewrite the manuscript in terms of methodology to describe Cryo-EM technique.

Reviewer #2: This paper by Richardson et al. describes the cryo- EM structure of adenosine monophosphate nucleosidase (Amn) from a pathogenic bacteria, Klebsiella pneumoniae, which causes hospital-acquired pneumonia and sepsis. Here are the points for revision.

Major points

This manuscript suffers from poor data analysis, the author says in the result, “The cryo-EM structure of K. pneumoniae Amn primarily differs from the apo homologs in its loop regions away from the active site (Figure 4A), either by lacking density for the loop entirely or being in a clearly different conformation”

-In case of lacking density, how the model was built for this region? I do not see any disconnect, missing segment, in Figure 4a.

Figure 4a does not reflect differences clearly, and a magnified view of the loop region would be needed to see the actual conformation differences.

At 3.1 Å resolution, the difference for side-chain residues should be clear. A picture showing side-chain confirmation would be helpful to reach this conclusion.

In catalytic activity, the Km and Vmax for the other two species were performed in different experiments, not at the same time and in the same setup. Reaching a conclusion may not be appropriate. Some unnecessary statement “though the --------- intervening decades”, is not required.

In figure 2B, I don’t see high-resolution features such as density for bulky side chain residues, particularly in a helix. That should be evident at this 3.1 Å resolution, and it looks overall like a low-resolution map. To gain readers’ confidence, detailed cryo-EM data processing information, including representative micrographs, 2D class, 3D class, and final post processed is required.

Minor point

In introduction; The acronym ESKAPE in the introduction needs expansion.

In result; the Figure 2 parts are not in order e.g. Figure 2C comes first in the text.

In result; ‘mixed secondary structure’ is not correct, needs to explain in terms of the secondary structure topology or domain. Same in Figure 2 legend also.

1T8S is a PDB id, it should be written as PDB ID; 1T8S, throughout the text.

In table 1; add one more column for the name of the organism for the source of the protein.

On searching EMD ID 26838 in EMDB shows no entry. Check this out.

Reviewer #3: The manuscript describes the structure of K. pneumoniae adenosine monophosphate nucleosidase determined by cryoelectron

microscopy and structure is compared with other known structures.

Overall manuscript is informative bur there are several concerns. The overall structure of the enzyme is not described, at least it should have been described briefly.

There are no images of EM, secondary classification, other than Figure 2B image. At least these should have been in supplementary data.

The authors describe structural differences in loops at such a low resolution. Are the lops well defines in EM structure?

Can they use this low resolution structure for drug design or inhibitor screening?

The manuscript does not describe or give the reference to the literature, explaining the importance of this enzyme. Does the importance of this enzyme shown in this organism?

Overall the authors should describe the manuscript with more information.

Reviewer #4: This paper reports the cryo-EM structure and catalytic activity of Adenosine Monophosphate Nucleosidase from Klebsiella Pneumoniae. However, the mauscript should be revised considering the following comments.

Comment 1: Page3

“Biochemical studies of nucleotide processing pathways in Azotobacter vinelandii and

Escherichia coli (E. coli) identified a key difference between prokaryotes and eukaryotes in their

regulation of the levels of adenosine monophosphate (AMP), a key component of both RNA

synthesis and energy storage.” Provide References.

Comment 2: Page 8

All PDB structures identified by BLAST as

close Amn homologs were aligned to Amn in UCSF Chimera (A)

Provide sequence comparison highlighting the conserved residues and discuss functionally significant residues.

Comment 3:

Page 8 line number 145:

“broader comparison of K. pneumoniae Amn to its homologs can be assessed by

sequence conservation”

Provide the phylogenetic relation between the available structures.

Comment 4:

Page 8 line number 149:

“the loop covering the active site is strongly conserved”

Provide the structural alignment of the active site residues.

Comment 5:

Page 10 line number 182

“while the crystal structure leads to significant underestimation of the flexibility of several N-terminal loops the catalytic core of the enzyme is effectively invariant between these species.”

Discuss any functional significance of the N-terminal loops in Amn, if not then there is no need to include this sentence.

Comment 6:

The discussion should be precise and elaborate and should only discuss their findings. The KpAmn structure should be compared with that of the other homologs in detail highlighting the functionally relevant substructures.

6. PLOS authors have the option to publish the peer review history of their article (what does this mean?). If published, this will include your full peer review and any attached files.

Reviewer #1: **Yes: **Dr. Ethayathulla Abdul samath

Reviewer #2: No

Reviewer #3: No

Reviewer #4: **Yes: **Pravindra Kumar

---

## [Author Response · Author response to Decision Letter 0]

12 Aug 2022

Our responses to reviewer comments are attached in a separate file.

---

## [Decision Letter · Decision Letter 1]

9 Sep 2022

Structure of Klebsiella Pneumoniae Adenosine Monophosphate Nucleosidase

PONE-D-22-14060R1

Dear Dr. French,

We’re pleased to inform you that your manuscript has been judged scientifically suitable for publication and will be formally accepted for publication once it meets all outstanding technical requirements.

Kind regards,

Janesh Kumar, Oh.D.

Academic Editor

PLOS ONE

Additional Editor Comments (optional):

Please remove the older version of Fig. 3 and provide PDB/EMDB deposition details in the final version before publication.

Reviewers' comments:

Reviewer's Responses to Questions

**Comments to the Author**

1. If the authors have adequately addressed your comments raised in a previous round of review and you feel that this manuscript is now acceptable for publication, you may indicate that here to bypass the “Comments to the Author” section, enter your conflict of interest statement in the “Confidential to Editor” section, and submit your "Accept" recommendation.

Reviewer #2: All comments have been addressed

Reviewer #3: All comments have been addressed

Reviewer #4: All comments have been addressed

2. Is the manuscript technically sound, and do the data support the conclusions?

Reviewer #2: Yes

Reviewer #3: Yes

Reviewer #4: Yes

3. Has the statistical analysis been performed appropriately and rigorously? 

Reviewer #2: Yes

Reviewer #3: N/A

Reviewer #4: Yes

4. Have the authors made all data underlying the findings in their manuscript fully available?

Reviewer #2: No

Reviewer #3: Yes

Reviewer #4: Yes

5. Is the manuscript presented in an intelligible fashion and written in standard English?

Reviewer #2: Yes

Reviewer #3: Yes

Reviewer #4: Yes

6. Review Comments to the Author

Reviewer #2: The authors have addressed all the points except not shared PDB ID and EMDB ID. Please make sure the coordinate and Cryo-EM maps are available for the scientific community immediately after publication.

Reviewer #3: Almost all of our questions and observations were addressed. The present version of the manuscript can be accepted.

Reviewer #4: Authors have submitted the revised manuscript on "Structure of Klebsiella Pneumoniae Adenosine Monophosphate Nucleosidase". They have answered all the queries and comments made by me.

7. PLOS authors have the option to publish the peer review history of their article (what does this mean?). If published, this will include your full peer review and any attached files.

Reviewer #2: No

Reviewer #3: No

Reviewer #4: No

---

## [Editor Report · Acceptance letter]

13 Oct 2022

PONE-D-22-14060R1 

Structure of *Klebsiella Pneumoniae* Adenosine Monophosphate Nucleosidase 

Dear Dr. French:

I'm pleased to inform you that your manuscript has been deemed suitable for publication in PLOS ONE. Congratulations! Your manuscript is now with our production department. 

Kind regards, 

on behalf of

Dr. Janesh Kumar 

Academic Editor

PLOS ONE